# UNSUPERVISED DOMAIN ADAPTATION WITH IMPUTATION

## ABSTRACT

Motivated by practical applications, we consider unsupervised domain adaptation for classification problems, in the presence of missing data in the target domain. More precisely, we focus on the case where there is a domain shift between source and target domains, while some components in the target data are systematically absent. We propose a way to impute non-stochastic missing data for a classification task by leveraging supervision from a complete source domain through domain adaptation. We introduce a single model performing joint domain adaptation, imputation and classification which is shown to perform well under various representative divergence families ($\mathcal{H}$-divergence, Optimal Transport). We perform experiments on two families of datasets: a classical digit classification benchmark commonly used in domain adaptation papers and real world digital advertising datasets, on which we evaluate our model's classification performance in an unsupervised setting. We analyze its behavior showing the benefit of explicitly imputing non-stochastic missing data jointly with domain adaptation.

## 1 INTRODUCTION

When dealing with machine learning applications in the real world, data usually come with several imperfections that make classical algorithms hardly deployable. One of these issues is that data are often incomplete. Typically, while capturing data coming from different locations with several sensors per location, a sensor may randomly fail or even may be just missing at a given location. Such a situation can also occur in disease diagnosis in multi-modal medical imaging where one of the modalities fails or is not available; for example the positron emission tomography (PET) modality which reveals metabolic information for clinical tests requires ingesting a radioactive tracer which poses health risks and is often missing (Cai et al., 2018). Similarly, in computational advertising applications, information is missing for users who do not have a prior history on a merchant's website while their global clicking behavior across websites may be known. Another common issue is that data used for training and deployment may differ in their generation process: data may be collected on different devices, background noise or compression schemes may affect differently training or deployment data, leading to a shift in data distribution. This has given rise to the important literature of Domain Adaptation (Pan & Yang, 2010; Kouw & Loog, 2019). These two issues are usually independently addressed by developing models handling only the missing data or the domain adaptation problem.

In this paper, motivated by practical advertising applications, we consider unsupervised domain adaptation (i.e. labels are not available in the target domain) for classification when (1) part of input data is missing in the target domain thus requiring some form of imputation, (2) there is no possible supervision in the target domain for imputation thus requiring indirect supervision from the source domain, and (3) there exists a domain shift between the source and target distributions requiring domain adaptation. More precisely we consider this adaptation-imputation setting for non-stochastic missing data, i.e. when the same features are missing for all target samples. This contrasts with many imputation problems which take benefit of stochasticity in missing features.

We propose a model that handles unsupervised domain shift and missing data assuming non-stochastic missing data in the target domain. The model learns to perform imputation for the target domain while aligning the distributions of the source and target domains in a latent space, thus going beyond the simple juxtaposition of a data imputation module followed by a domain-invariant feature

representation learning module. Imputation makes use of an indirect supervision from the complete source domain. This key property allows us to handle non-stochastic missing data, while satisfying the constraints related to adaptation and to the classification objective. The imputation process plays several roles in our global architecture as it provides us with information about the missing data for the target domain while contributing to the domain-invariant loss and the reconstruction loss. Extensive empirical evidence on handwritten digits and Click-Through-Rate prediction (CTR) domain adaptation problems illustrate the benefit of the proposed model.

The original contributions are the following:

- We introduce a new problem : joint unsupervised domain adaptation and imputation for classification motivated by practical applications;

- We propose a new model for handling the problem end-to-end. It learns to generate relevant missing information while aligning source and target distributions in a latent space and to classify source instances;

- We evaluate the model not only on academic benchmarks but also on challenging real world advertising data.

## 2 RELATED WORK

We review below typical related work for domain adaptation and data imputation.

### 2.1 UNSUPERVISED DOMAIN ADAPTATION (UDA)

A number of shallow learning methods approach Domain Adaptation by weighting individual observations during training. These methods focus either on data importance-weighting (Cortes & Mohri, 2014; Zadrozny, 2014) or on class importance-weighting (Z. Lipton & Smola, 2018). Recent deep learning methods try to align the distributions of the two domains, for example by embedding them in a joint latent space. There are two main directions for learning joint embeddings. One is based on adversarial training, making use of GAN extensions. The other one directly exploits explicit distance measures between distributions such as Wasserstein or Maximum Mean Discrepancy (MMD). For the former, the seminal work of Ganin & Lempitsky (2015) learns to map source and target domains onto a common embedding space, by optimizing a double objective: on the one hand they minimize an approximation of the $\mathcal{H}$-divergence between the source and target embeddings via adversarial training, on the other hand they learn to classify the source data embeddings. This influential work has been followed by several extensions and variants. ADDA (Tzeng et al., 2017) advocates the use of two different mappings for the source and the target domains based on the argument that this is more suitable when the marginals are different in the two domains. Liu & Tuzel (2016) trains coupled generative adversarial network (CoGAN) for learning a joint distribution of multi-domain images, that can be used for UDA. Bousmalis et al. (2017) use a generator to map the source to the target domain while training the classifier on the learned representations using source labels. CDAN (Long et al., 2017) improves the domain discriminator by conditioning it on classifier predictions.

A second family of approaches proposes metric based divergences such as MMD (Long et al., 2015) for measuring the loss between source and target representations. DeepJDOT (Damodaran & Kellenberger, 2018) makes use of an optimal transport formulation to align the joint distributions in a latent space. In addition to feature alignment they perform label distribution alignment following Courty et al. (2017). All these works rely on the assumption of covariate shift and consider that full input data is available for both source and target domains. Our two models (`ADV` and `OT`) can be seen respectively as extensions of Ganin & Lempitsky (2015) and Damodaran & Kellenberger (2018) for the missing data problem.

### 2.2 IMPUTATION

Data imputation is a classical problem addressed by several methods (Little & Rubin, 2002; Van Buuren, 2018; Murray, 2018). The usual setting is different from ours since it considers reconstructing the whole missing data in the input space, while we consider 1) reconstruction in a latent space and 2) partial reconstruction since we are interested in the information relevant to the classification task

only. Recent generative models like GANs (Goodfellow et al., 2014) or VAEs (Kingma & Welling, 2013; Rezende et al., 2014) have been adapted for data imputation in Yoon et al. (2018) and Mattei & Frellsen (2019) respectively. GAIN (Yoon et al., 2018) is an extension of conditional GANs where the generator takes as input an incomplete data and performs imputation while the discriminator is trained to guess for each sample if each variable is original or imputed. Mattei & Frellsen (2019) suggests a method based on deep latent variable models and importance sampling that offers tighter likelihood bound compared to the standard VAE bound. Most approaches consider a supervised setting where 1) paired complete and incomplete data are available and 2) missingness corresponds to a stochastic process (e.g. a mask distribution for tabular data), 3) imputation is performed in the original feature space. Note that this is different from our setting where there is no direct supervision (supervision is only provided indirectly through the source domain) and missingness is non-stochastic which makes the problem harder since one cannot compute statistics on different incomplete samples. The general approach with generative models is to learn a distribution over imputed data which is similar to the one of plain data. This comes in many different instances and usually, generative training alone is not sufficient; additional loss terms are often used. In paired problems, i.e. when each missing datum is associated to a plain version of the datum, these additional terms consist of a reconstruction term imposed by a MSE contraint (Isola et al., 2016b). In unpaired problems a cycle-consistency loss is imposed as in Zhu et al. (2017). Li et al. (2019); Pajot et al. (2019) are among the very few approaches addressing unsupervised imputation in which full instances are never directly used. Both extend the AmbientGAN (Bora et al., 2018) framework and consider stochastic missingness.

Our imputation problem is closer to the ones addressed in some forms of inpainting or for multi-modality missing data. The former problem is addressed e.g. in Pathak et al. (2016) who proposes an encoder-decoder model trained according to a joint reconstruction and adversarial loss. The latter is addressed in Cai et al. (2018) who considers the case of multi-modality when one or more modalities are systematically absent, but they do not consider adaptation. They propose to learn to reconstruct the missing modality distribution conditionally to the observed one. Both approaches are fully supervised. Ding et al. (2014) is the only paper we are aware of that considers imputation as we do. Their approach is based on low rank constraints and dictionary learning to guide the transfer between domains. We do not use this method as a baseline due to the complexity and running time of this method which relies on singular value decompositions and dictionnary learning.

## 3 PROBLEM DEFINITION

Let us denote respectively $(x_S, y_S) \in \mathbb{R}^n \times \mathbb{R}$ and $(x_T, y_T) \in \mathbb{R}^n \times \mathbb{R}$, data from the source and target domains where $x_-$ is an input, $y_-$ the associated label and $n$ is the dimension of the input space. "$-$" holds for either source $S$ or target $T$. The joint distribution on each domain is denoted respectively $p_S(X, Y)$ and $p_T(X, Y)$. We consider that $x_-$ has two components, $x_- = (x_{-_1}, x_{-_2})$. The problem we address is Unsupervised Domain Adaptation (UDA) with missing features in the target domain. More precisely, we make the following hypotheses. **Missingness:** We assume that features are the same across domains and that source features $x_S = (x_{S_1}, x_{S_2})$ are always available while in the target domain only $x_{T_1}$ is available and $x_{T_2}$ is systematically missing. For advertising applications for example, $x$ would characterize the user browsing behaviour on merchant sites; $x_{-_1}$ characterizes global user features aggregated over his navigation history, which are known for all users; $x_{-_2}$ characterizes user history on a target merchant site. Source domain would consist of all users who already visited this merchant site and target domain of users who never visited this site. **UDA:** we assume that source labels $y_S$ are available whereas target labels $y_T$ are unknown. **Covariate shift:** we assume covariate shift as in most UDA papers e.g. Ganin & Lempitsky (2015).

## 4 ADAPTATION-IMPUTATION MODEL

As in many generative approaches to UDA, the objective is to project source and target data onto a common latent space in which data distributions from the two domains match, and to learn a classifier on the source data that performs well on the target domain. The novelty of our approach is to offer a solution to deal with datasets in which some information, $x_{T_2}$, is systematically missing in the target domain. Our model, denoted `Adaptation-Imputation`, performs three operations jointly: imputation of missing information for the target data, alignment of the distributions

of source and target, and classification of source instances. The three operations are performed in a joint embedding space and all the model's components are trained together. The term imputation is used here in a broad sense: our goal is not to recover the whole missing $x_{T_2}$, but to recover information from $x_{T_2}$ that will be useful for adaptation and for the target data classification objective. This is achieved via a generative model, which for a given datum in the target domain and conditionally on the available information $x_{T_1}$, attempts to generate the required missing information. Because $x_{T_2}$ is systematically missing for target data, there is no possible supervision in the target domain; instead we use distant supervision from the source data while transferring to the target domain. We consider two variants of the same model based on different divergence measures between source and target distributions: the Wasserstein distance and the $\mathcal{H}$-divergence approximated through adversarial training. For simplicity we describe in the main text the adversarial version `ADV` and defer the Optimal Transport `OT` description to Appendix B. We report the results obtained with both models in Section 5.

## 4.1 TRAINING

Our model is composed of three different modules responsible for adaptation, imputation and classification, that share parameters and are trained in parallel. For simplicity, we describe each component in turn, but it should be reminded that they all interact and that their parameters are all optimized according to the three objectives mentioned above. The interaction is discussed after the individual module descriptions. The model's components are illustrated in Figure 1 (a).

**Adaptation**  The latent space representations of source and target domains are denoted with a tilde notation: $\widetilde{x}_S = (\widetilde{x}_{S_1}, \widetilde{x}_{S_2})$ and $\widetilde{x}_T = (\widetilde{x}_{T_1}, \widetilde{x}_{T_2})$. Referring to Figure 1 (a), $\widetilde{x}_{-_1} = g_1(x_{-_1})$, ($\widetilde{x}_{-_1}$ denotes either $\widetilde{x}_{S_1}$ or $\widetilde{x}_{T_1}$) is the mapping of the observed component $x_{-_1}$ onto the latent space and $\widetilde{x}_{-_2} = h \circ g_1(x_{-_1})$ is the second component's latent representation generated from $x_{-_1}$. This generation mechanism will be described later. Adaptation aligns the distributions $(\widetilde{x}_{S_1}, \widetilde{x}_{S_2})$ and $(\widetilde{x}_{T_1}, \widetilde{x}_{T_2})$ in the latent space. For the `ADV` model, alignment is performed via a classical adversarial loss operating on the latent representations:

$$L_1 = E_{x \sim p_S(X)} \log D_1(\widetilde{x}_S) + E_{x \sim p_T(X)} \log(1 - D_1(\widetilde{x}_T)) \tag{1}$$

where $D_1(\widetilde{x})$ represents the probability that $\widetilde{x}$ comes from the source rather than the target.

**Imputation**  Imputation amounts at generating an encoding $\widetilde{x}_{T_2}$, in the latent space, for the missing information in the target data, conditioned on the available information $x_{T_1}$. Our objective here is to generate missing information which is relevant for the classification objective. Since we never have access to any target component $x_{T_2}$, we learn to perform imputation based on the source data. More precisely, we learn to generate $\widetilde{x}_{S_2}$ from $x_{S_1}$ through the generator $h$, $\widetilde{x}_{S_2} = h \circ g_1(x_{S_1})$, as depicted in Figure 1. We want $h$ to generate the missing information $\widetilde{x}_{S_2}$ associated to the observed $x_{S_1}$. For that we perform two operations in parallel. First, we align the distribution of $\widetilde{x}_{S_2}$ with the distribution of $\widehat{x}_{S_2} = g_2(x_{S_2})$, that is a direct mapping of $x_{S_2}$ onto the shared latent space, using an adversarial loss described below. The intuition is that both $g_1$ and $g_2$ are simple mappings operating respectively on $x_{S_1}$ and $x_{S_2}$ while $h$ acts as a generator conditioned on $x_{S_1}$ for generating $\widetilde{x}_{S_2}$. Moreover, we not only impose this distribution alignment, but would also like $\widetilde{x}_{S_2}$ to represent missing information relative to $x_{S_2}$ and associated to a specific $x_{S_1}$. For that, we use a reconstruction term in parallel to the above alignment, in our case a MSE distance between $\widetilde{x}_{S_2}$ and $\widehat{x}_{S_2}$. This MSE term guarantees that the imputed $\widetilde{x}_{S_2}$ truly represents information present in $x_{S_2}$. Similar ideas combine distribution matching and MSE conditioning and have been used e.g. in Isola et al. (2016a); Pathak et al. (2016). The learned mappings are used to perform imputation on the target data $\widetilde{x}_{T_2} = h \circ g_1(x_{T_1})$.

The imputation loss has thus two components. The first is the adversarial term $L_{ADV}$ responsible for aligning $\widetilde{x}_{S_2}$ and $\widehat{x}_{S_2}$, $L_{ADV} = E_{x_2 \sim p_S(X_2)} \log D_2(\widehat{x}_{S_2}) + E_{x_1 \sim p_S(X_1)} \log(1 - D_2(\widetilde{x}_{S_2}))$. The second is the reconstruction term $L_{MSE} = E_{x \sim p_S(X)} \|\widetilde{x}_{S_2} - \widehat{x}_{S_2}\|_2^2$. The total imputation loss is then:

$$L_2 = \lambda_{ADV} \times L_{ADV} + \lambda_{MSE} \times L_{MSE} \tag{2}$$

where $\lambda_{ADV}, \lambda_{MSE}$ are hyperparameters. The two processes of imputation and adaptation influence each other. Both are also influenced by the classification process described below. Its effect

on imputation is to force the generated $\widetilde{x}_{S_2}$ to contain information about $x_{S_2}$ relevant for the classification task. This information is transferred via adaptation to the target domain when generating $\widetilde{x}_{T_2}$.

**Classification**  The last component of the model is a classifier $f$, trained on the source domain mapping $\widetilde{x}_S$ for the classification task as classically done for UDA. The corresponding loss is:

$$L_3 = E_{x \sim p_S(X)} L_{Disc}(f(\widetilde{x}_S), y_S) \tag{3}$$

where $L_{Disc}$ is typically a cross-entropy loss.

**Overall loss**  The overall loss function $L$ will be the weighted sum of the adaptation, imputation and classification losses:

$$L = \lambda_1 \times L_1 + \lambda_2 \times L_2 + \lambda_3 \times L_3 \tag{4}$$

where $\lambda_1, \lambda_2, \lambda_3$ are hyperparameters and the final optimization problem is:

$$\min_{g_1, g_2, h, f} \max_{D_1, D_2} L \tag{5}$$

**Interaction between the model's components**  Both mappings $g_1, g_2$ and generator $h$ appear in the three terms of the loss function in Equation 4, meaning that they should learn to perform the three tasks simultaneously. $g_1$ learns to map the $x_{S_1}$ and $x_{T_1}$ components onto the latent space, the mappings being denoted respectively $\widetilde{x}_{S_1}$ and $\widetilde{x}_{T_1}$. $h$ learns to generate missing information $\widetilde{x}_{T_2}$ from $\widetilde{x}_{T_1}$. The formed $\widetilde{x}_-$ is generated such that it fulfills the classification objective. $g_2$ on its side should fulfill the imputation objective while preserving part of the information present in $x_{S_2}$. Note that our model makes use of a unique mapping $g_1$ for both source and target domains. Separate mappings could have been used for the two domains, but the proposed solution was found to be more robust and to reduce the number of parameters during learning.

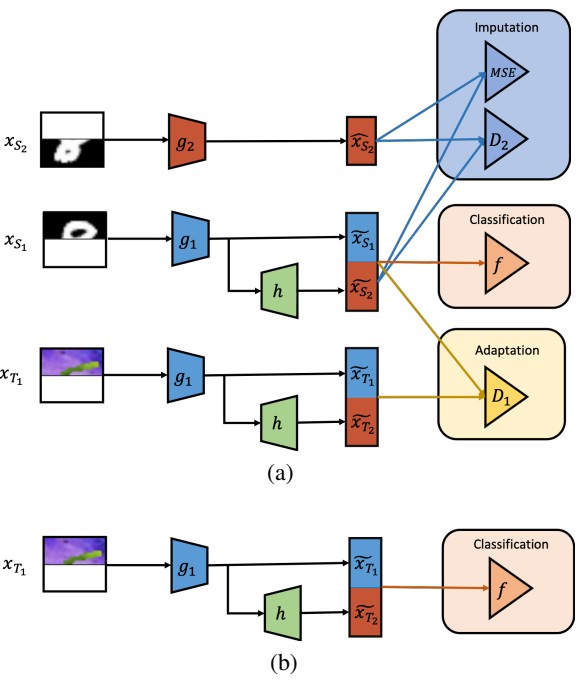

Figure 1: `Adaptation-Imputation` model: (a) training, (b) inference.

**Implementation**  Let us now detail the implementation of this model. For adversarial training, discriminators $D_1$ (adaptation) and $D_2$ (imputation) will be implemented by binary classifiers. $D_1$ is trained to distinguish between source $\widetilde{x}_S$ and target $\widetilde{x}_T$ mappings while $D_2$ is trained to separate

imputed $\widetilde{x}_{S_2}$, generated from $x_{S_1}$, from $\widehat{x}_{S_2}$ a direct embedding of $x_{S_2}$. We use the gradient reversal trick in Ganin & Lempitsky (2015) for implementing the min-max condition and define two gradient reversal networks on $D_1$ and $D_2$. We use an adaptive update of the scale of the gradients in $D_1$ and $D_2$ and optimize $L_1$, $L_2$ and $L_3$ jointly as synthesized in Algorithm 1 in the Appendix. In practise we fix all hyperparameters but $\lambda_{MSE}$ to 1, additional tuning could yield improved performance.

## 4.2 INFERENCE

At inference, given $x_{T_1}$, we generate $\widetilde{x}_T = (\widetilde{x}_{T_1}, \widetilde{x}_{T_2})$ with $\widetilde{x}_{T_1} = g_1(x_{T_1})$ an embedding of $x_{T_1}$ and a generated $\widetilde{x}_{T_2}$, encoding part of the missing information $x_{T_2}$ in $x_T$, as illustrated in Figure 1 (b). We use for the latter the following mapping: $\widetilde{x}_{T_2} = h \circ g_1(x_{T_1})$ where $g_1$ is as above and $h$ is the generative mapping conditioned on $\widetilde{x}_{T_1}$. Finally $\widetilde{x}_T$ is used as input to the classifier $f$.

## 5 EXPERIMENTS

### 5.1 DATASETS AND EXPERIMENTAL SETTING

**Datasets**   Experiments are performed on two types of datasets. The first one is a classical digits classification benchmark used in many domain adaptation studies which we will refer to as `digits` and transformed to fit our missing data setting. The second one corresponds to advertising datasets. The task here is binary classification: one wants to predict Click-Through-Rate (CTR) or Conversion Rate (CR) given user behavior. This is one of the problem that has initially motivated our adaptation-imputation framework. We use two such datasets: `ads-kaggle` is a public kaggle dataset[1], while `ads-real` has been gathered internally and corresponds to real advertising traffic. Further details on datasets and preprocessing are presented in Sections 5.2, 5.3 and in Appendix C.

**Baselines**   We report results for the following models:

- Full models: `Source-Full` is trained without adaptation on the full $x_S$ and tested on full $x_T$ when the latter is available (`digits`); `Adaptation-Full` adds adaptation to this model.
- Missing models: `Source-Missing` and `Adaptation-Missing` do the same but considering full $x_S$ while $x_T$ is incomplete: $x_T = (x_{T_1}, 0)$, i.e. $x_{T_2}$ is set to 0.
- Partial models: `Source-Partial` and `Adaptation-Partial` is a variant of the above setting where only the first component $x_{-_1}$ for source and target are considered for adaptation and classification while the second ones $x_{-_2}$ are simply ignored.
- Imputation models: `Adaptation-Imputation` corresponds to our model.
- Naive model: `Naive` is used for `ads-kaggle` to provide a reference loss value for this dataset. It predicts for all examples the mean CTR value as computed using source training data only.

`Adaptation-Full` is an upper bound of the performance of `Adaptation-Imputation` since it uses full information while $x_{T_2}$ is not available in practice. `Adaptation-Missing` and `Adaptation-Partial` can be considered as lower bounds for our model since they only perform adaptation and no imputation.

Parameters and architecture of the neural networks used for the different models and experiments are presented in Appendix D. Hyperparameters are chosen using the Deep Embedded Validation estimator introduced in You et al. (2019) combined with heuristics and typical UDA values. Further details are given in Appendix D.2.1.

We present the results for `digits` and `ads` respectively in Sections 5.2, 5.3. Section 5.4 presents ablation studies. Reported results are mean value and standard deviation over five different initializations; best results are indicated in **bold**.

### 5.2 DIGITS

**Description**   For `digits`, we consider the unsupervised adaptation between two datasets among MNIST (LeCun et al. (1998)), USPS (Hull (1994)), SVHN (Netzer et al. (2011)) and MNIST-M

---

[1]http://labs.criteo.com/2014/02/kaggle-display-advertising-challenge-dataset/

(Ganin & Lempitsky (2015)). The direction MNIST → SVHN is not considered as the task is difficult even for traditional UDA (Ganin & Lempitsky, 2015). All tasks are 10-class classification problems. From the complete image datasets, we build datasets with missing input values.

**Half digit missing** In a first series of experiments, we removed one half of each image, the horizontal bottom part. We report classification accuracy in Table 1 for the different adaptation problems and models (ADV and OT). Removing half of the image leads to a strong performance decrease for `Source-Partial` and `Source-Missing` with respect to the upper bounds provided by `Source-Full`, respectively between 10 and 20 points of accuracy, and between 15 and 45 points. This is partially recovered when training with adaptation (`Adaptation- Partial`, `Adaptation-Missing`, for both ADV or OT). But the gap is still important with respect to the upper bound, i.e. `Adaptation-Full`. In all cases, `Adaptation-Imputation` clearly increases the performance; between 10 and 25 points of accuracy over `Adaptation-Missing` and 2 to 20 points over `Adaptation-Partial`. This is a very significant improvement which validates the importance of the imputation component. In Section 5.4 we show that the simultaneous use of imputation and adaptation is required for reaching this level of performance. Imputation or adaptation alone are well behind the jointly trained instance of the model. However, it does not reach the upper bound performance of `Adaptation-Full` where the difference lies between 10 and 25 accuracy points. Moreorever, `Adaptation-Imputation` beats the non-adapted `Source-Full` baseline on several datasets. Both the ADV and OT versions exhibit the same general behavior. In the reported results in Table 1, ADV performance is higher than OT. This is because performance is highly dependent on the NN architectures and we tuned our NNs for ADV. OT models may reach performance similar to ADV but we find that it requires models with an order of magnitude more parameters. To keep the comparison fair, we thus used the same NN models for both ADV and OT. Imputation models achieve their highest performance when the adaptation task between domains is complex (MNIST → MNIST-M, SVHN → MNIST) illustrating the importance of imputation when transfer is difficult. In all experiments, the performance of `--Partial` model where "−" refers to `Source` or `Adaptation`, are usually higher than the `--Missing` model. Our understanding is that setting missing components to zero tends to increase distance between source and target distributions, compared to just ignoring them, making the classification and adaptation problems harder.

Table 1: Classification accuracy performance in % on `digits` for the two training criteria on the target domain test set. Standard deviation is in %.

| | MNIST → USPS | | USPS → MNIST | | SVHN → MNIST | | MNIST → MNIST-M | |
|---|---|---|---|---|---|---|---|---|
| Method | ADV | OT | ADV | OT | ADV | OT | ADV | OT |
| Source-Full | 71.5±2.7 | | 74.2±2.7 | | 58.1±1.1 | | 28.3±1.4 | |
| Adaptation-Full | 88.3±2.4 | 92.6±1.7 | 95.0±0.4 | 93.9±0.6 | 77.6±3.5 | 76.1±1.4 | 77.2±4.9 | 46.9±3.9 |
| Source-Missing | 25.7±3.7 | | 39.2±2.6 | | 31.5±2. | | 14.4±1.1 | |
| Adaptation-Missing | 48.4±4.8 | 60.9±6.3 | 67.5±2.2 | 65.3±5.2 | 47.1±5.7 | 37.5±6.2 | 34.7±2.5 | 20.2±2.5 |
| Source-Partial | 52.9±9.7 | | 54.3±1.6 | | 44.6±1.9 | | 19.1±2.6 | |
| Adaptation-Partial | 71.5±3.2 | 64.0±5.0 | 80.0±1.4 | 72.0±1.8 | 45.5±1.9 | 47.9±1.8 | 29.4±1.6 | 26.8±4.4 |
| Adaptation-Imputation | **75.2±1.5** | **66.8±1.3** | **81.5±0.8** | **72.5±2.7** | **54.1±1.4** | **49.2±1.5** | **58.5±1.6** | **29.2±1.4** |

**Missing patch size** In a second group of experiments on `digits`, we analyze the evolution of the performance of the models with respect to the size of the missing information in the target domain. For that, we vary the size of the missing patch removing a percentage of the image $p$ with $p \in \{30\%, 40\%, 50\%, 60\%, 70\%\}$ on SVHN → MNIST for ADV models, keeping the same hyperparameters as the ones used for $p = 50\%$. We report the mean values over five runs in Figure 2. We notice that `Adaptation-Imputation` constantly beats the other baselines regardless of the missing patch size. The figure exhibits borderline cases when the size of the missing patch be-

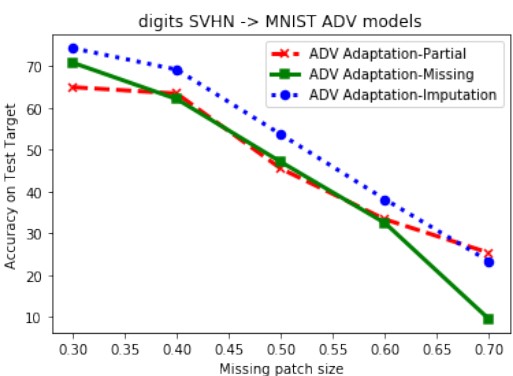

Figure 2: Missing patch size study

comes very small ($< 30\%$) or very large ($> 65\%$). When the missing patch is too small most of the information for predicting the target label is already available thus simple models perform already well; while when it becomes too big, too few information is available to guarantee efficient reconstructions from the non-missing patch.

## 5.3 ADS

**Description**   We performed a second series of tests on two advertising datasets: `ads-kaggle` and `ads-real`. The `ads` datasets correspond to binary classification problems; the task is to predict the probability that a **user** exposed to an ad from a **target partner** (e.g. Booking) on a **publisher** (e.g. NY Times) will click (`ads-kaggle`) or make a purchase (`ads-real`) conditioned on the user history. A row in the dataset corresponds to a display i.e. an ad opportunity of a click or purchase for a given (user, partner) pair at a given time on a given publisher site. The source domain is composed of users who already had interactions with a target partner. The target domain is composed of users with no history on a target partner. We consider all partners in a given traffic. For the two domains, $x_{-_1}$ features correspond to aggregated user features on all the partners, while $x_{-_2}$ corresponds to user - target partner specific interaction which is known for the source domain but unknown for the target domain. Note that besides missingness, there is also an adaptation problem since statistics for new users are usually different from those of known users (e.g. in terms of frequency of a partner's website visits) as seen in Appendix E. In real datasets, traffic in the source domain is usually abundant while scarce in the target domain. Statistics for each dataset are provided in Table 5 in the Appendix; exact preprocessing used is provided in Appendix C.

**Results**   For this group of experiments, we report the results only for `ADV` models since the trend has been observed to be similar on `digits` for both `ADV` and `OT`. For `ads` datasets, missing features do not exist, so we do not report the `--Full` models' results on these datasets. The classes being imbalanced, accuracy is not relevant here so we report another performance measure, cross-entropy (CE) between the predicted values and the true labels on the target domain which is considered as the most reliable metric to estimate revenue. Note that given the test set size of `ads-kaggle`, an improvement of 0.001 in logloss is considered as practically significant (Wang et al., 2017). For the ads problem and for large user bases, a small improvement in prediction accuracy can lead to a large increase in a company's revenue. For all experiments, we report in Table 2 CE on target test for `ads-kaggle` and `ads-real`.

A first observation is that `Adaptation-Imputation` is significantly better than the baselines on both datasets (Table 2). For `ads-kaggle` it improves by 2.3% the best adaptation model (`Adaptation-Missing`) while for `ads-real` the improvement reaches 6.3% over the best second which happens to be `Source-Partial`. A second observation is that for any model, adaptation consistently

Table 2: CE on `ads` for `ADV` models

| Dataset | ads-kaggle | ads-real |
|---|---|---|
| Naive | 0.403 | X |
| Source-Missing | 0.545±0.019 | 0.663±0.011 |
| Source-Partial | 0.406±0.00046 | 0.622±0.0048 |
| Adaptation-Missing | 0.397±0.0057 | 0.660±0.025 |
| Adaptation-Partial | 0.403±0.0030 | 0.634±0.0082 |
| Adaptation-Imputation | **0.389±0.014** | **0.583±0.013** |

improves over the same model without adaptation. The only exception is the `--Partial` setting in `ads-real`. A third observation is that the missing component indeed contains relevant information: CE performance on source data (not reported in Table 2) shows that `Source-Missing` which exploits the $x_{-_2}$ component is consistently higher than `Source-Partial` which does not exploit this component, leading to relative gains of the former over the latter of 5.6% on `ads-kaggle` and 8.2% on `ads-real`. `Adaptation-Imputation` is able to generate and to exploit this information.

## 5.4 ABLATION ANALYSIS

We analyze now the role and importance of the different components of our model, and compare with the results from Tables 1 and 2. We perform experimentation on the public datasets, `digits` and `ads-kaggle` and report results in Table 3 and Figure 3.

**Importance of adaptation**  We compare the performance of the model with and without the adaptation term $L_1$ in Equation 4. When removing adaptation, inference is performed as before, by feeding $\widetilde{x}_T = (\widetilde{x}_{T_1}, \widetilde{x}_{T_2})$ to the classifer $f$. This means that we only rely on the imputation and classification losses to learn the parameters of the model. Results appear on the top of Table 3. For all datasets, the adaptation component considerably increases the performance, from 10 to 30 points for `digits` and by a significant 0.009 CE value on `ads-kaggle`.

**Imputation mechanism**  Imputation, cf. Equation 2, combines adversarial training (`ADV`) and conditioning on the input datum via the MSE loss (`MSE`). The objective is to learn from $x_{S_1}$ $\widetilde{x}_{S_1} = g_1(x_{S_1})$ and to generate missing information in $x_{S_2}$, $\widetilde{x}_{S_2} = h(\widetilde{x}_{S_1})$. `ADV` aligns the distributions of $\widetilde{x}_{S_2}$ and $\widehat{x}_{S_2}$ while `MSE` can be thought as performing some form of regression. For a given partial information $x_{S_1}$, there are possibly several potential $x_{S_2}$ and thus $\widetilde{x}_{S_2}$. `ADV` allows to focus on a specific mode of $\widehat{x}_{S_2}$, while `MSE` will favour a mean value of the distribution. Results on Table 3, second group of rows, show that for `digits`, combining the two influences (`MSE` and `ADV`) leads to improved results compared to using separately each loss. `MSE` alone already provides good performance, while using only `ADV` is clearly below. For this classification task, identifying the most relevant mode improves the performance over simple regression ($L_{MSE}$). Note that reconstruction is an ill posed problem since the task is inherently ambiguous - different digits may be reconstructed from one half of an image. We performed tests with a stochastic input component in order to recover different modes, but the performance was broadly similar. Achieving diversity with Conditional GANs remains an open research topic (Yang et al., 2019).

For the `ads-kaggle` dataset, the performance of `MSE` and `MSE + ADV` are similar. This is analyzed deeper in an additional series of experiments with several weighted combinations of `MSE` and `ADV`. Results are provided in Table 3 third group of rows, for both `digits` and `ads-kaggle` and are plotted for `ads-kaggle` in Figure 3. For `digits`, this confirms that the equal weights selected for our experiments are indeed generally a good choice reducing the burden of hyperparameter selection, while for `ads-kaggle` performance could be slightly improved with other weightings. One can see on Figure 3 that `ADV` induces a high variance in the results (left part of x-axis) while `MSE` stabilizes the performance (right part of x-axis). The former allows for better maximum performance but with high variance: performance ranges from 0.35 to 0.7 on the target domain. A small contribution from `MSE` (here $\lambda_{MSE} = 0.005$) stabilizes the results.

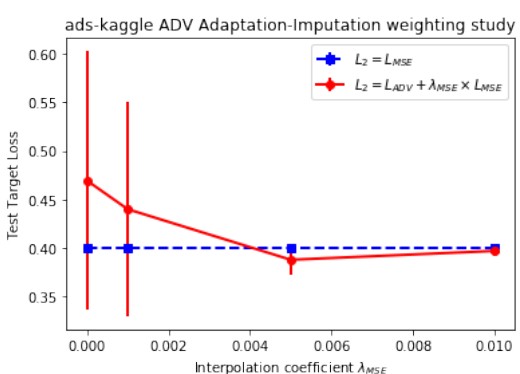

Figure 3: `ADV-MSE` weighting on `ads-kaggle`

Table 3: Accuracy on `digits` and CE on `ads-kaggle` for `ADV Adaptation-Imputation`

| Dataset | digits | | | | ads-kaggle |
|---|---|---|---|---|---|
| Adaptation direction | MNIST → USPS | USPS → MNIST | SVHN → MNIST | MNIST → MNIST-M | Known → New |
| $L_2 + L_3$ | 64.2±1.8 | 51.3±2.5 | 44.5±1.4 | 24.1±2.6 | 0.410±0.0020 |
| $L_1 + L_2 + L_3$ | **75.2±1.5** | **81.5±0.8** | **54.0±1.4** | **58.5±1.6** | **0.401±0.0014** |
| $L_{MSE}$ | 71.9±3.7 | 81.4±1.2 | 52.5±3.7 | 56.5±2.8 | **0.400±0.0014** |
| $L_{ADV}$ | 28.6±3.2 | 39.4±5.2 | 28.8±3.8 | 30.0±3.7 | 0.469±0.13 |
| $L_{ADV} + L_{MSE}$ | **75.2±1.5** | **81.5±0.8** | **54.0±1.4** | **58.5±1.6** | 0.401±0.0014 |
| $0.1 \times L_{ADV} + L_{MSE}$ | 73.4±2.7 | 81.3±0.8 | 53.0±2.0 | 56.2±2.6 | 0.401±0.0021 |
| $L_{ADV} + 0.001 \times L_{MSE}$ | 37.3±2.5 | 31.2±3.8 | 45.0±2.6 | 50.0±3.4 | 0.440±0.11 |
| $L_{ADV} + 0.005 \times L_{MSE}$ | 47.8±3.7 | 49.6±5.8 | 46.0±2.6 | 50.6±2.2 | **0.388±0.015** |
| $L_{ADV} + 0.01 \times L_{MSE}$ | 53.6±2.4 | 57.0±3.6 | 43.4±1.1 | 51.0±2.5 | 0.397±0.0046 |
| $L_{ADV} + 0.1 \times L_{MSE}$ | 68.2±4.2 | 50.3±6.8 | 54.0±2.1 | 51.5±3.6 | 0.402±0.0046 |
| $L_{ADV} + L_{MSE}$ | **75.2±1.5** | **81.5±0.8** | **54.0±1.4** | **58.5±1.6** | 0.401±0.0014 |

## 6 Conclusion

We have proposed a new model to solve unsupervised adaptation problems in the presence of non-stochastic noise in the target domain, by using distant supervision from a complete source domain

through domain adaptation and imputing missing values on the target domain in a latent space. This method uses only labelled source instances and leads to important gains on classical adaptation benchmarks over baseline models for two representative families of divergences (optimal transport, adversarial training). We have demonstrated on real world advertising datasets that these methods can be used for problems with missing features in advertising. Potential follow-ups include: extending this method to a semi or fully supervised setting on the target domain; considering simultaneously domain and target shift which frequently occurs in real world problems while still being an open problem; introducing increased diversity in the generation process.

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

## A    PSEUDO-CODE

**Input:** $N$: number of epochs, $k$: batch size, $lr_i$: initial learning rate
Initialize parameters of $g_1, g_2, D_1, D_2, f, h$
**while** $n_{epochs} < N$ **do**
    Sample $\{x_S{}^i, y_S{}^i\}_{1 \le i \le k}$ from $p_S$
    Sample $\{x_{T_1}^j\}_{1 \le j \le k}$ from $p_T$
    Decay learning rate and update gradient scale at each batch
    Compute $L = L_1 + L_2 + L_3$ performing joint adaptation, imputation, classification
    Update $D_1, D_2$ by ascending $L$ through Gradient Reversal Layer
    Update $f, g_1, g_2, h$ by descending $L$
**end**

**Algorithm 1:** Training procedure for `ADV` Adaptation-Imputation

## B    `OT` FORMULATION

We present here in more details `Adaptation-Imputation` using Optimal Transport as a divergence metric. The formulation is slightly different compared to `ADV` models.

### B.1    FORMULATION OF THE IMPUTATION `OT` MODEL

We replace the $\mathcal{H}$-divergence approximation given by the discriminators $D_1$ and $D_2$ by the Wasserstein distance between source and target instances ($D_1$) and true and imputed feature representations ($D_2$), following the original ideas in Shen et al. (2018); Damodaran & Kellenberger (2018). In practice, we compute the Wasserstein distance using its primal form by finding a joint coupling matrix $\gamma$, using a linear programming approach (Peyré et al., 2019).

**Adaptation**    In Damodaran & Kellenberger (2018); Courty et al. (2017), the optimal transportation problem is formulated on the joint $(X, Y)$ distributions. Similarly to Shen et al. (2018), in our case, we focus on a plan that acts only on the feature space without taking care of the labels. This leads to the following optimization problem :

$$L_1 = \min_{f, \boldsymbol{\gamma_1}, g} \sum_{ij} \left( ||\widetilde{x}_{S_{1_i}} - \widetilde{x}_{T_{1_j}}||^2 + ||\widetilde{x}_{S_{2_i}} - \widetilde{x}_{T_{2_j}}||^2 \right) \boldsymbol{\gamma_1}_{ij} \tag{6}$$

where $\boldsymbol{\gamma_1}_{ij}$ is the alignment value between source instance $i$ and target instance $j$.

**Imputation**    For the imputation part, we keep the reconstruction MSE component in Equation 2 and derive the distribution matching loss as:

$$L_{OT} = \min_{f, \boldsymbol{\gamma_2}, g} \sum_{ij} ||\widetilde{x}_{S_{2_i}} - \widehat{x}_{S_{2_j}}||^2 \boldsymbol{\gamma_2}_{ij} \tag{7}$$

where $\boldsymbol{\gamma_2}_{ij}$ is the alignment value between source instance $i$ and $j$.

The final imputation loss is:

$$L_2 = \lambda_{OT} \times L_{OT} + \lambda_{MSE} \times L_{MSE} \tag{8}$$

**Classification**    The classification term in Equation 3 is unchanged.

### B.2    IMPLEMENTATION

The optimization problem in Equation 5 is solved in two stages following an alternate optimization strategy:

- we fix all parameters but $\gamma_1$ and $\gamma_2$ and find the joint coupling matrices $\gamma_1$ and $\gamma_2$ using EMD

- we fix $\gamma_1$ and $\gamma_2$ and solve $\min_{g_1, g_2, h, f} L$

In practice, we first minimize $L_3$ for a couple of epochs (taken to be 10 for `digits`) then minimize $L_1 + L_2 + L_3$ in the remaining epochs.

Learning rate and parameters are explained further in Section D.

## C  DATASET DESCRIPTION

### C.1  DIGITS

Table 4: Statistics on `digits` datasets

|  | USPS | MNIST | SVHN | MNIST-M |
|---|---|---|---|---|
| Train | 7438 | 60k | 73257 | 60k |
| Test | 1860 | 10k | 26032 | 10k |
| Size | $28 \times 28$ | $28 \times 28$ | $32 \times 32$ | $28 \times 28$ |
| Channels | 1 | 1 | 3 | 3 |

**Pre-processing**  We scale all images to be $32 \times 32$ and we normalize the input in $[-1, 1]$. When adaptation involves a domain with three channels (SVHN or MNIST-M) and a domain with a single channel, we simply triplicate the channel of the latter domain. As in Damodaran & Kellenberger (2018) we use balanced source batches which proves to increase performance especially when the source dataset is imbalanced (e.g. SVHN and USPS datasets) while the target dataset (usually MNIST derived) is balanced. Scaling the input images enables us to use the same architecture across datasets. In practise the embedding size is 2048 after preprocessing. For missing versions, we set pixel values to zero in a given patch. Figure 4 shows MNIST-M digits with varying missing patch size.

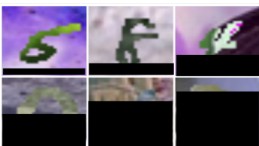

Figure 4: missing-features MNIST-M

**Evaluation**  The `digits` datasets are provided with a predefined train / test split. We report accuracy results on the target test set and use the source test set as validation set (Section D.2.1). The number of instances in each dataset is reported in Table 4. We run each model five times with a different seed for initialization (using Xavier Uniform initialization) and report mean and standard deviations in Table 1.

**Infrastructure**  Experiments are run using a Tesla V100-SXM2 GPU machine with 32GB RAM. Code is written in PyTorch `https://pytorch.org/` (Paszke et al., 2017).

### C.2  ADS

**Preprocessing**  Table 5 lists various statistics on the traffic for the two `ads` datasets; we describe in this paragraph how datasets are processed.

**`ads-kaggle`**  The Criteo Kaggle dataset is a reference dataset for CTR prediction and groups one week of log data. The objective is to model the probability that a user will click on a given ad given his navigation context: $p(Click|Context)$. Positives refer to displays which are clicked

and negatives to displays not clicked. For each datum, there are 13 continuous and 26 categorical features. We divide the traffic into two domains using feature number 30; for a given value for this categorical feature, all instances have a single missing numeric feature (feature number 5). We then construct an artificial dataset simulating transfer between known and new users. We process the original Criteo Kaggle dataset to have an equal number of source and target data. We then perform train / test split on this dataset keeping 20% of data for testing. We used in our experiments the continuous features; to show the benefit of modelling additional missing features, we extend the missing features list to features 1, 5, 6, 7, 11 and 12 by setting them to zero on the target domain. After these operations, 6 features are missing and 7 are non-missing. Preprocessing consists in normalizing continuous features using a log transform.

**ads-real** The data collected in this dataset is similar to `ads-kaggle`. We filter out non-clicks and the final task is to model the sale probability for a clicked ad: $p(Sale|Click = 1, Context)$. Positives refer to clicked ads which lead to a sale; negatives to clicked ads which did not lead to a sale. We use one week of sampled logs as a training set, and use the following day data as the test set. This train / test definition is used so as to better correlate with the performance of a model used online. We use typical features in production models on this dataset and focus on a sample of continuous features aggregated across user timelines describing the clicking and purchase behavior of a user. In comparison to `ads-kaggle` more continuous features are used. The count features can be User-centric i.e. describe the global activity of the user (number of clicks, displays, sales done globally across partners) or User-partner features i.e. describing the history of the user on the given partner (number of clicks, sales... on the partner). The latter are missing for new users. Counts are aggregated across varying windows of time and categories of partner catalog. We bucketize these count features using log transforms and consider these as categorical features by one-hot encoding the different buckets based on the vocabulary file. After processing the final input size is 596 with 29 features used. 12 features are missing and 17 are non-missing.

Table 5: Statistics on `ads` datasets

| Dataset | ads-kaggle | | | | ads-real | | | |
|---|---|---|---|---|---|---|---|---|
| Domain | Source | | Target | | Source | | Target | |
| Split | Train | Test | Train | Test | Train | Test | Train | Test |
| Positive | 246 872 | 61 841 | 92 333 | 22 943 | X | X | X | X |
| Negative | 699 621 | 174 783 | 854 160 | 213 681 | X | X | X | X |
| Total | 946 493 | 236 624 | 946 493 | 236 624 | 24 465 756 | 3 760 233 | 819 073 | 147 358 |
| $P(Y)$ | 0.2608 | 0.2613 | 0.0976 | 0.0970 | X | X | X | X |

**Evaluation** On both datasets the train and test sets are fixed. We run each model five times with a different seed for initialization (using Xavier Uniform initialization) and report mean and standard deviations in Table 2.

**Infrastructure** For `ads-real`, models are trained using internal large-scale distributed infrastructure. For `ads-kaggle`, we keep the same setting and code as in `digits`.

## D IMPLEMENTATION DETAILS

### D.1 NEURAL NET ARCHITECTURE

#### D.1.1 DIGITS

We experiment with the `ADV` and `OT` versions of our imputation model. For `ADV` models, we use the DANN model description in Ganin & Lempitsky (2015); for `OT` we use the DeepJDOT model description in Damodaran & Kellenberger (2018). Both models can be considered as simplified instances of our corresponding `ADV` and `OT` imputation models when no imputation is performed.

Performance of the adaptation models is highly dependent on the NN architectures used for adaptation and classification. In order to perform fair comparisons and since our goal is to evaluate the potential of joint adaptation-imputation-classification, we selected these architectures through pre-

liminary tests and use them for both the `ADV` and `OT` models. The two models are described below and illustrated in Figure 5.

- Feature extractors $g_1$ and $g_2$ consists of three convolutional layers with $5 \times 5$ kernel and 64 filters interleaved with max pooling layers with a stride of 2 and $2 \times 2$ kernel. The final layer has 128 filters. We use batch norm on convolutional layers and ReLU as an activation after max pooling. As in Damodaran & Kellenberger (2018) we find that adding a sigmoid activation layer as final activation is helpful.

- Classifier $f$ consists of two fully connected layers with 100 neurons with batch norm and ReLU activation followed by the final softmax layer. We add Dropout as an activation for the first layer of the classifier.

- Discriminator $D_1$ and $D_2$ is a single layer NN with 100 neurons, batch norm and ReLU followed by the final softmax layer. On USPS $\rightarrow$ MNIST and MNIST $\rightarrow$ USPS dataset we use a stronger discriminator network which consists of two fully connected layers with 512 neurons.

- Generator $h$ consists of two fully connected layers with 512 neurons, batch norm and ReLU activation. This architecture is used for `ADV` and `OT` imputation models. In practice using wider and deeper networks increases classification performance with the more complicated classification tasks (SVHN $\rightarrow$ MNIST, MNIST $\rightarrow$ MNIST-M); in these cases we add an additional fully connected network with 512 neurons. The final activation function is a sigmoid.

We use the same architecture described above for full, missing and partial models to guarantee fair comparison. As a side note, the input to the imputation model's classifier is twice bigger as in the standard adaptation models.

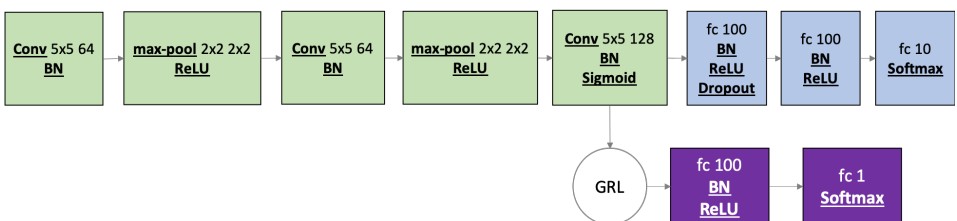

Figure 5: Base architecture for the `ADV` DANN model

### D.1.2 ADS

We experiment with `ADV` models only. As input data is numeric and low dimensional, architectures are simpler than in `digits`.

**ads-kaggle** Our feature extractor is a three layered NN with 128 neurons and with a final sigmoid activation. The classifier is taken to be a single layered NN with 128 neurons and a final softmax layer. Activations are taken to be ReLUs. The domain discriminator is taken to be a two layered NN with 128 neurons and a final softmax layer. Finally the reconstructor is taken to be a two-layered NN with 256 neurons and final sigmoid activation.

**ads-real** Input features after processing are fed directly into the feature extractors $g_1$, $g_2$ consisting of two fully connected layers with 128 neurons. The classifier and discriminator is taken to be single-layered NN with 25 neurons. The reconstructor is taken to be a two-layered NN with 128 neurons. Inner activations are taken to be ReLUs and the final activation of the feature extractor is taken to be a sigmoid.

### D.2 NETWORK PARAMETERS

### D.2.1 HYPERPARAMETER TUNING

Tuning hyperparameters for UDA is tricky as we do not have access to the label in the target distribution and thus cannot choose parameters minimizing the target risk on a validation set. Several papers

set hyperparameters through reverse cross-validation (Ganin & Lempitsky, 2015). Other approaches developed for model selection are based on risk surrogates obtained by estimating an approximation of the risk value on the source based on the similarity of source and target distributions (without the labels). In the experiments, we used a recent estimator, Deep Embedded Validation (DEV) (You et al., 2019) for tuning the initial learning rate; $\lambda_1$ and $\lambda_{OT}$ in `OT`. For other parameters, we used heuristics and typical hyperparameter values from UDA papers (such as batch size) without further tuning. We use a cross entropy link function on the source validation set; this value provides a proxy for the target test risk. Using parameters from the original paper, this estimator helps select parameter ranges which perform reasonably well. We keep the estimator unchanged for the full and missing adaptation models. In the imputation case the discriminator used for computing importance sampling weights discriminates between $[\widetilde{x}_{S_1}, \widetilde{x}_{S_2}]$ and $[\widetilde{x}_{T_1}, \widetilde{x}_{T_2}]$ (cf. $D_1$ in Figure 1).

### D.2.2 DIGITS

We find that the results are highly dependent on the NN architecture and the training parameter setting. In order to evaluate the gain obtained with `Adaptation-Imputation`, we use the same NN architecture for all models (`ADV` and `OT`) but fine tune the learning rates for each model using the DEV estimator (other parameters do not have a significant impact on the classification performance).

**ADV**  We use an adaptive approach as in Ganin & Lempitsky (2015) for decaying the learning rate $lr$ and updating the gradient's scale $s$ between 0 and 1 for the domain discriminators. We choose the decay values used in Ganin & Lempitsky (2015) ie. $s = \dfrac{2}{1 + \exp(-10 \times p)} - 1$ and $lr = \dfrac{lr_i}{(1 + 10 \times p)^{0.75}}$ where $p$ is ratio of current batches processed over the total number of batches to be processed without further tuning. We tune the initial learning rate $lr_i$, chosen in the range $\{10^{-2}, 10^{-2.5}, 10^{-3}, 10^{-3.5}, 10^{-4}\}$ following Section D.2.1. In practise we take $lr_i = 10^{-2}$ for `ADV Adaptation-Imputation`, `Adaptation-Full`, `Adaptation-Partial` and $lr_i = 10^{-2.5}$ for `ADV Adaptation-Missing`. We use Adam (Kingma & Ba, 2014) as the optimizer with momentum parameters $\beta_1 = 0.8$ and $\beta_2 = 0.999$ and use the same decay strategy and initial learning rate for all components (feature extractor, classifier, reconstructor). Batch size is chosen to be 128; we see in practise that initializing the adaptation models with a source model with smaller batch size (such as 32) can be beneficial.

**OT**  We choose parameter $\lambda_1 = \lambda_{OT} = 0.1$ in Equations 4 and 8 after tuning in the range $\{10^{-1}, 10^{-2}, 10^{-3}\}$ using DEV; the goal is indeed to down weight the contribution of alignment through Wasserstein distance which should be scaled compared to the classification term. Following Damodaran & Kellenberger (2018), batch size is taken to be 500 and we use EMD as Optimal Transport metric. We initialize adaptation models with a source model in the first 10 epochs and divide the initial learning rate by two as adaptation starts for non-imputation models. For `Adaptation-Imputation` we follow a decaying strategy on the learning rate and on the adaptation weight as explained in the next item. We choose $lr_i$ in the range $\{10^{-2}, 10^{-2.5}, 10^{-3}, 10^{-3.5}, 10^{-4}\}$. In practise we fix $lr_i = 10^{-2}$ for all models.

**Imputation parameters**  In practice we can define a weight for each term in Equation 2 and 4. Further studies are conducted in Section 5.4 on weightings in Equation 2: in `digits` experiments we choose $L_2 = L_{MSE} + L_{ADV}$ for ADV and `OT` to reduce the burden of additional feature tuning. For `ADV` model, we fix parameters in Equation 4 to 1. In the `OT` model, we vary $\lambda_1$ between 0 and 0.1 and $\lambda_2$ between 0 and 1 following the same schedule as the gradient scale update for `ADV` models. Although not necessary to achieve good mean values, this proves to reduce significantly the variance.

### D.2.3 ADS

We use an adaptive strategy for updating the gradient scale and the learning rate with the same parameters as in the `digits` dataset. Optimizer is taken to be Adam. Batch size is taken to be big so that target batches include sufficient positive instances.

**ads-kaggle** The initial learning rate is chosen in the range $\{10^{-4}, 10^{-5}, 10^{-6}, 10^{-7}\}$ using DEV and fixed to be $10^{-6}$ for all models. Batch size is taken to be 500 and we initialize models with a simple classification loss for five epochs. We run models for 50 epochs after which we notice that models reach a plateau. We find that adding a weighted MSE term allows to achieve higher stability (as measured by variance) as further studied in Section 5.4. In a similar fashion to Pathak et al. (2016), we tune this weight in the range $\{1, 10^{-1}, 10^{-2}, 7.5 \times 10^{-3}, 5 \times 10^{-3}, 10^{-3}\}$. We find that 0.005 offers the best compromise between mean loss and variance. Moreover on this dataset we use a faster decaying strategy for the discriminator's $D_2$ and the reconstructor's $h$ learning rate, $lr = \dfrac{lr_i}{(1 + 30 \times p)^{0.75}}$ to achieve higher stability in the training curves while the feature extractor $g_1$, $g_2$ and $D_1$'s learning rate are unchanged.

**ads-real** The initial learning rate is chosen in the range $\{10^{-4}, 10^{-5}, 10^{-6}\}$ and fixed to be $10^{-6}$ for all models. The learning rate is decayed with the same parameters as digits for all models. We run models for ten epochs which provides a good trade-off between learning time and classification performance. Batch size is taken to be 500. We choose $L_2 = L_{MSE} + L_{ADV}$ without further tuning; this simple weight achieves already good results.

## E  VISUALIZATION OF FEATURES' DISTRIBUTION ON ADS-KAGGLE

The goal of this section is to visualize the domain shift existing in ads-kaggle pre-adaptation to show that the problem comprises both a domain shift and an imputation problem. Table 6 reports mean and standard deviation on each feature's value over the domain and Figure 6 plots the histogram of the normalized distributions for better visualization. The $y$-axis is not normalized and corresponds to real counts. We see that feature 5 is missing on the target domain and that distributions are different in shape, mean and variance on each domain.

Table 6: Mean and standard deviation for each feature on ads-kaggle

| Dataset | ads-kaggle | |
| --- | --- | --- |
| Domain | Source | Target |
| feature 1 | 0.80±2.21 | $4.4 \times 10^{-4}$±0.041 |
| feature 2 | 9.16±13.04 | 9.01±13.42 |
| feature 3 | 4.40±6.32 | 3.44±6.19 |
| feature 4 | 2.58±3.27 | 0.94±2.31 |
| feature 5 | 61.09±37.67 | 0.0±0.0 |
| feature 6 | 11.26±12.24 | 0.090±1.69 |
| feature 7 | 4.10±6.23 | 0.0034±0.13 |
| feature 8 | 5.12±4.50 | 1.91±4.26 |
| feature 9 | 14.32±11.57 | 3.273±5.36 |
| feature 10 | 0.046±0.22 | $1.35 \times 10^{-5}$±0.0037 |
| feature 11 | 1.08±2.11 | $4.25 \times 10^{-4}$±0.029 |
| feature 12 | 0.083±0.78 | $6.68 \times 10^{-5}$±0.018 |
| feature 13 | 2.74±3.59 | 1.21±3.36 |

## F  EMBEDDING VISUALIZATION ON DIGITS

In this section we visualize the embeddings learned by the various models on digits by projecting the embeddings in a 2D space using t-SNE (the original embedding size being 2048). Figure 7 represents the embeddings learned for ADV models on MNIST → MNIST-M. Figures 8 and 9 represent these embeddings for OT models respectively on MNIST → MNIST-M and MNIST → USPS.

On these figures, we see that imputation models generate feature representations that overlap better between source and target examples per class than the adaptation counterparts (although Adaptation-Partial does a good job at overlapping feature representations). This correlates with the accuracy performance on the test set. Moreover we notice, as expected, that missing and partial adaptation networks perform badly compared to the full adaptation counterparts which justifies the use of imputation model when confronted to missing data.

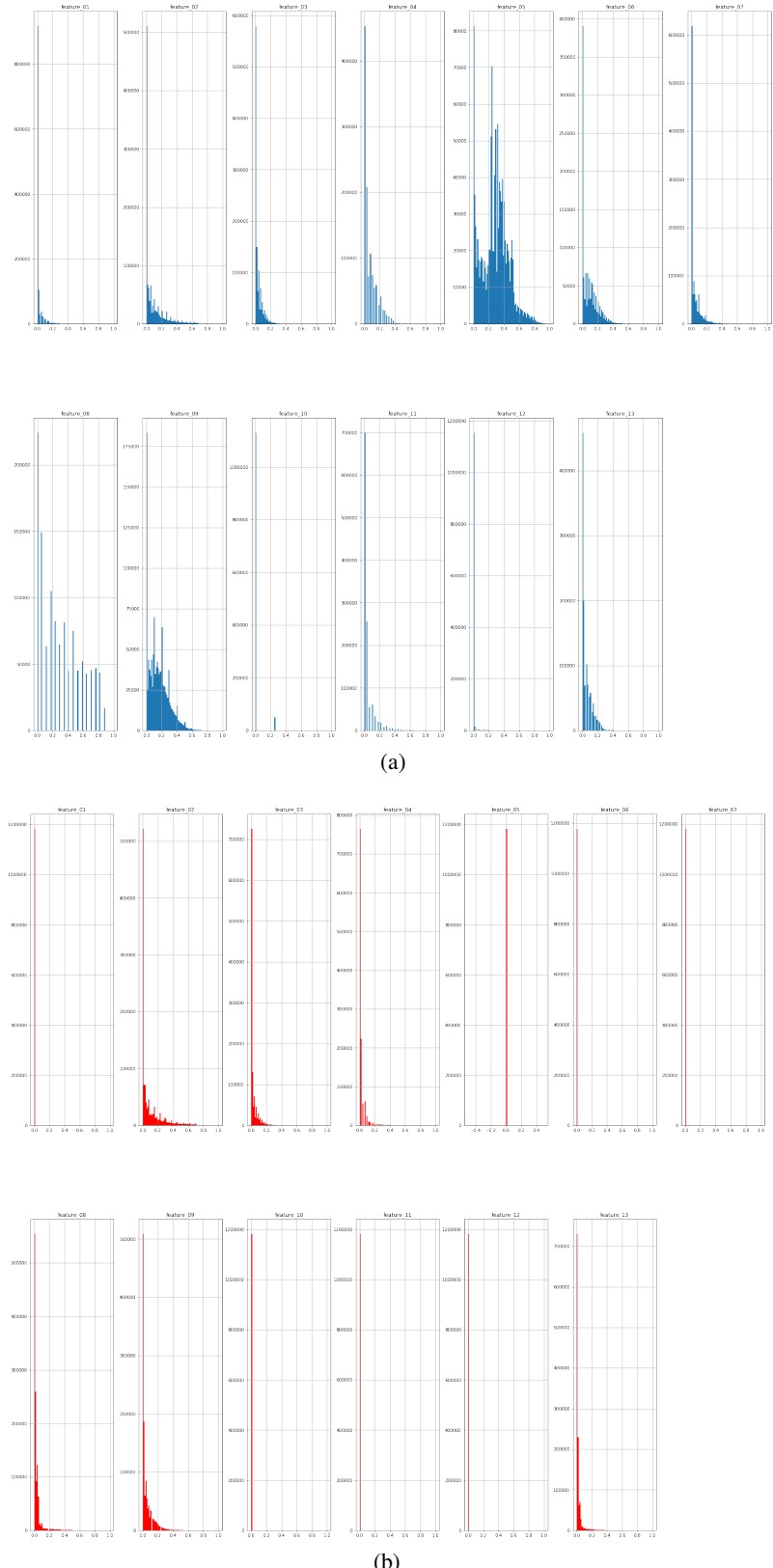

Figure 6: Features' normalized distribution for each input dimension (features 1 to 12): (a) Source domain (blue), (b) Target domain (red)

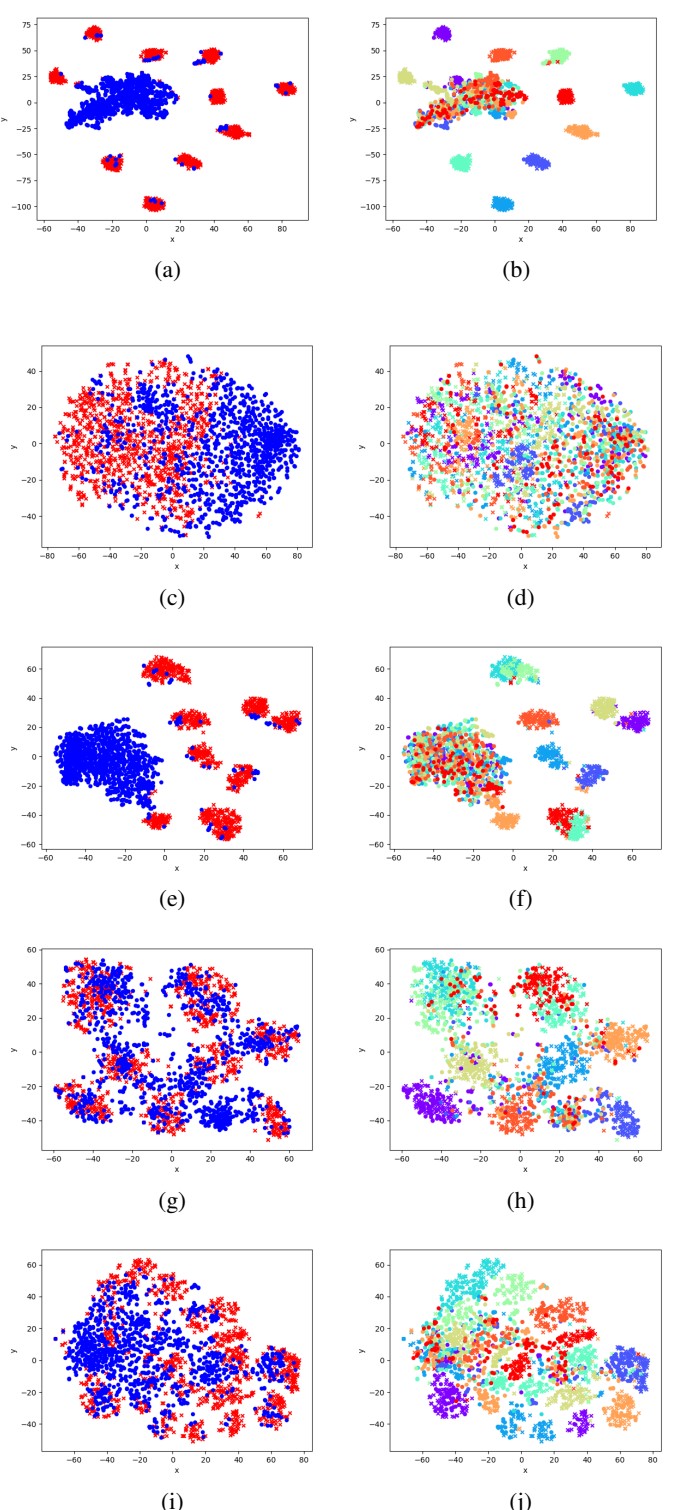

Figure 7: Embeddings for MNIST → MNIST-M dataset on a batch, for source missing (acc 14.4%) (a) (b); `ADV` missing (acc 32.6%) (c) (d); `ADV` partial (acc 27.0%) (e) (f) `ADV` with imputation (acc 56.3%); (g) (h) and `ADV` full (acc 74.2%) (i) (j). (a) (c) (e) (g) (i) represent the target (blue) and source (red) clusters, (b) (d) (f) (h) (j) represent the classes on source and target instances.

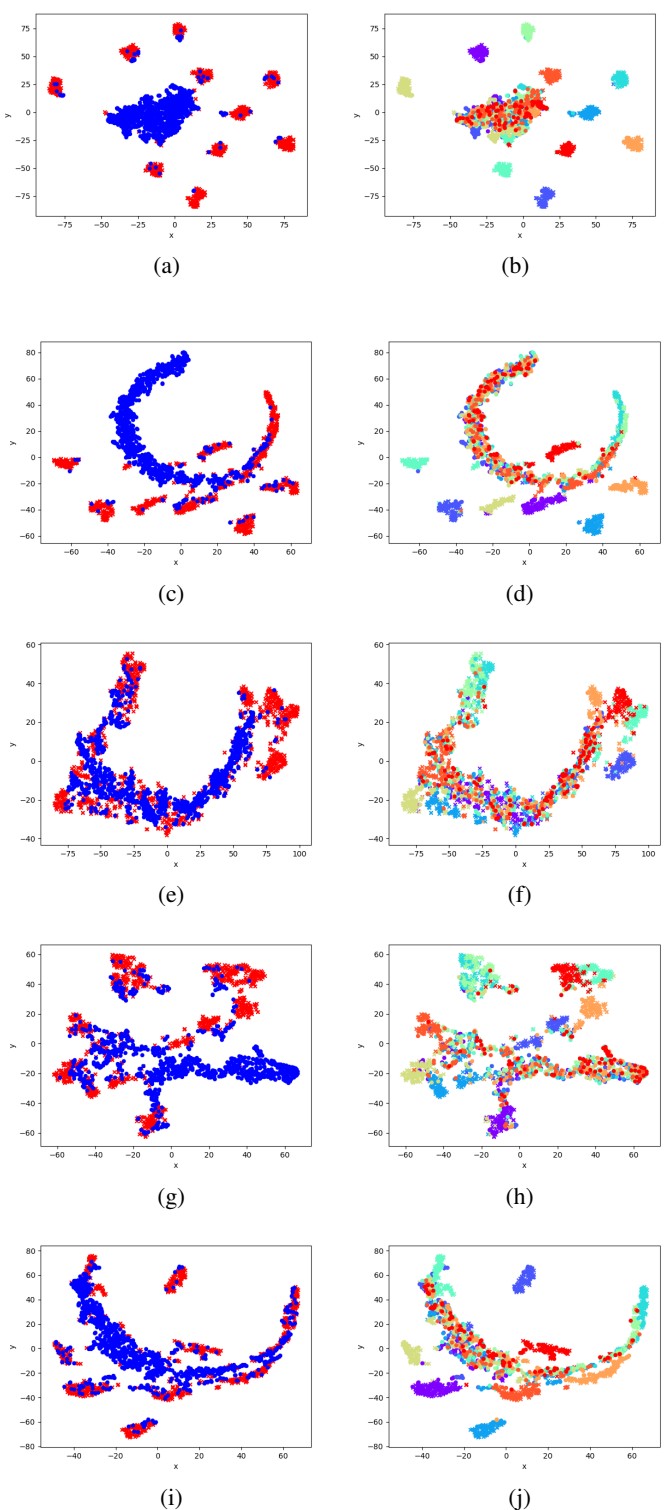

Figure 8: Embeddings for MNIST → MNIST-M dataset on a batch, for source missing (acc 14.5%) (a) (b); OT missing (acc 18.8%) (c) (d); OT partial (acc 27.75%) (e) (f); OT with imputation (acc 28.6%) (g) (h) and OT full (acc 45.9%) (i) (j). (a) (c) (e) (g) (i) represent the target (blue) and source (red) clusters, (b) (d) (f) (h) (j) represent the classes on source and target instances.

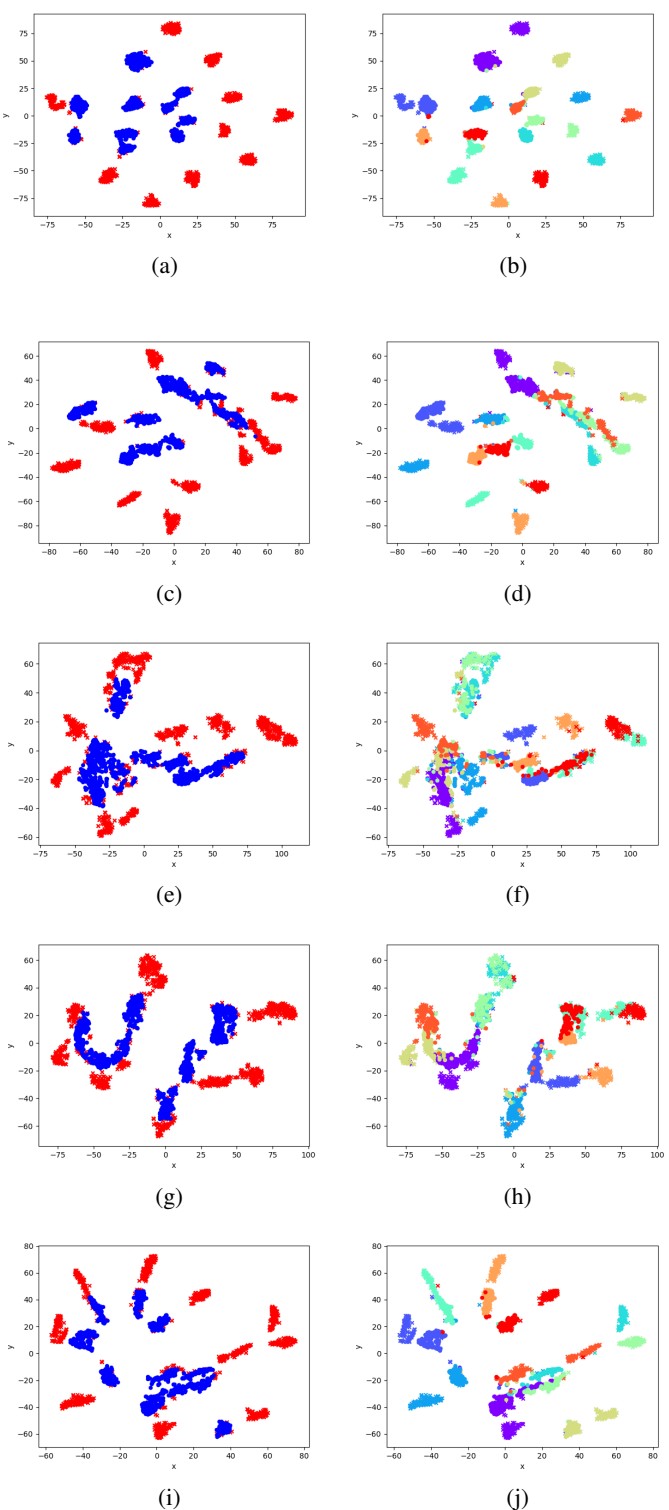

Figure 9: Embeddings for MNIST → USPS dataset on a batch, for source missing (acc 25.0%) (a) (b); `OT` missing (acc 58.0%) (c) (d); `OT` partial (acc 62.42%) (e) (f); `OT` with imputation (acc 65.2%) (g) (h) and `OT` full (acc 91.5%) (i) (j). (a) (c) (e) (g) (i) represent the target (blue) and source (red) clusters, (b) (d) (f) (h) (j) represent the classes on source and target instances.

