# OpenReview forum: "Unsupervised domain adaptation with imputation"
_ICLR.cc/2020/Conference — Reject_

### Official Review · AnonReviewer3 · 2019-10-22
**Official Blind Review #3**

**Rating:** 8

**Review:**

The submission describes an approach for unsupervised domain adaptation in a setting where some parts of the target data are missing.

Both UDA approaches as well as data completion approaches have a sizable research history, as laid out in the related work section (Section 5). The novelty here comes from the properties that a) domain adaptation and data imputation are handled in a joint manner, b) the missing data in the target domain is non-stochastic, and c) imputation is performed in a latent space. This maps to a fairly specific, but realistic enough set of real-world problems; the authors give an image recognition as well as an advertising prediction related problem as experimental examples.

The submission is overall well written and easy to understand. I'd rate the novelty as medium (smart combination of existing methods), but the exemplary experimental evaluation elevates it to more than a systems paper.

The method is described clearly in Section 3, and the joint training makes sense. I notice that not all hyperparameters ({lambda_adv, lambda_mse}, {lambda_1, lambda_2, lambda_3}) are truly needed. lambda_adv and lambda_1 could be canonically set to 1 for such a loss minimization problem, so why are the extraneous parameters included?

In addition to Section 3, the experimental evaluation on two very different data sets in Section 4 is highly detailed and describes the insights clearly, both qualitatively and quantitatively. I'm happy that mean standard deviations are reported on an acceptable experiment sample set size.
Regarding the different approaches: I'm wondering whether the higher performance of the ADV approach over OT (or the parameter hunger of OT over ADV) is only due to the tuning of the network architectures, or whether this is due to the approximations described in B.1.
The ablation study in Section 4.4 is interesting w.r.t. the trade-off it shows between stable, consistent, "average" results from an MSE loss term, vs. high-variance (and on average better) results when a choice of mode is forced using an adversarial loss term.

Minor comments:
- In Table 2, I am not sure what the first row ('Naive') refers to. As far as I can tell, it is not referenced in the text.
- I would move Section 5 (related work) to right after the introduction, as is common in conference papers and makes for smoother reading.
- Section 5.2: type "impainting" -> "inpainting"
- Appendix, section 'Pre-processing': It seems to me that there is a clear assumption made that the target set is balanced, since training happens with a balanced source set. Is this realistic in practical scenarios? There is work on DA with unequal class distributions between domains.

In summary, I can clearly recommend this submission for publication.

**Experience Assessment:**

I have published one or two papers in this area.

**Review Assessment: Checking Correctness Of Derivations And Theory:**

I carefully checked the derivations and theory.

**Review Assessment: Checking Correctness Of Experiments:**

I assessed the sensibility of the experiments.

**Review Assessment: Thoroughness In Paper Reading:**

I read the paper at least twice and used my best judgement in assessing the paper.

---

> ### Author Response · Authors · 2019-11-13
> **Answer Reviewer 3 - Replied to comments**
>
> We gratefully thank the reviewer for acknowledging the effort put in the experimental evaluation, for pointing out typos and advise on the structure of the paper.
>
> * Indeed as the reviewer notices, not all hyperparameters are used in practise in the experimental section. During initial tests, we experimented with different values of the hyperparameters and finally only tuned $\lambda_{MSE}$ in the experiments. We kept the initial formulation to indicate that the other parameters could also be tuned further and might yield improved classification performance. We have added a sentence clarifying this in the revised version in Section 4.1.
>
> * OT vs ADV: in preliminary tests, we figured out that the NN architectures used for mapping data onto the latent space had usually a much higher influence on the performance than the choice of the alignment method itself. In order to provide a fair comparison and also because our goal was to show the effectiveness of the proposed mechanism rather than reaching the best possible performance, we decided to use NN architectures with similar complexity for both models. The OT models used in [Damodaran2018] indeed require an order of magnitude more parameters than the ADV models [Ganin2015] to reach similar performance. Concerning the approximation, the OT model in [Damodaran2018] uses an alignment on the joint (X,Y) distributions while we are only aligning the X distributions. Hence, our approach is equivalent to a primal Wasserstein version of the domain adaptation method proposed in [Shen 2018]. In our tests, aligning on the joint distribution did not perform better than aligning only on the marginal. Then, our most plausible explanation is still that the difference of performance is due to tuning.
>
> * The Naive model is referenced in the Baselines paragraph (in the revised version it figures in Section 5.1) . This model refers to the likelihood computed using as a prediction the mean CTR on the training set and is a typical baseline used on advertising problems.
>
> * We have moved the related works to Section 2 in the revised paper upon the reviewer's recommendation and corrected the typo.
>
> * Indeed there is in practise no guarantee that the target domain will be balanced and usually it is not. However, dealing with both domain shift and label shift is more complex and requires additional mechanisms. Most work up to now only considered one of the two hypothesis (either domain or label shift). Recent work such as [Zhao2019], [Wu2019] has started to examine more complex settings and it is of course of strong practical interest. We have also started analyzing joint domain shift and unequal class distributions.
>
> [Ganin2015] Yaroslav Ganin et al.  Unsupervised Domain Adaptation by Backpropagation. 2015
> [Damodaran2018] Bharath Bhushan Damodaran et al. DeepJDOT : Deep Joint Distribution Optimal Transport for Unsupervised Domain Adaptation. 2018
> [Zhao2019] Han Zhao et al.  On learning invariant representation for domain adaptation. ICML 2019
> [Wu2019] Yifan  Wu et al.   Domain  adaptation  with asymmetrically-relaxed distribution alignment. ICML 2019
> [Shen2018] Shen Ji et al. Wasserstein Distance Guided Representation Learning for Domain Adaptation, AAAI 2018

---

### Official Review · AnonReviewer4 · 2019-10-23
**Official Blind Review #4**

**Rating:** 3

**Review:**

*Summary.* The paper presents and addresses the problem of performing domain adaptation when the target domain is systematically (i.e., not the result of a stochastic process) missing subsets of the data. The issue is motivated by applications where one modality of data becomes unavailable in the target domain (e.g., when deciding which ads to serve to new users, the predictor may have access to behavior across other websites but not on a specific merchant's website). The proposed method learns to map source and target data to a latent space where the representations for the source and target are aligned, the missing components of the target can be inferred, and classification can be performed successfully. These are achieved by adversarial/optimal transport loss on source and target features, a mean-squared error and adversarial loss on latent generation/imputation, and a cross entropy loss on source label prediction, respectively. Experiments are performed on digits and click-through rate (CTR) prediction and include a thorough set of baselines/oracles for comparison.

*Review.* While the problem statement is novel, I am unconvinced that the advertising experiment includes both a domain adaptation and imputation problem. I describe this in detail below. For this reason, I am giving the paper a weak reject.

*Questions that impacted rating.*
1. Ads experiment: From my understanding, the source domain is the traffic of users who have interacted with (clicked through to?) a specific partner and the target domain is the traffic of the users who have not interacted with that specific partner. The data that needs to be imputed is the click through rate for target users with that specific partner. In this case, it is not obvious to me why there is a domain shift between these two groups of users. This would imply that the traffic of source users and target users is different for other partners. I don't see why this would need to be true. Could the authors provide an explanation as to why this is the case (e.g., by showing that CTRs differ with other (partner, publisher) pairs between source and target). From my understanding, Table 5 only shows CTR averaged across all users in each domain, but does not show that the CTRs differ between source and target users for contexts/(partner, publisher) pairs (i.e., the results in table 5 could be due to the fact that the prior distribution over context is different for source and target users).

*Additional notes. Immaterial to rating.*
1. I personally felt that the motivation for UDA vs imputation in the first paragraph was a bit muddled. I think sticking to one example would make the motivation more clear to the reader. E.g., explain the prediction problem for medical imaging (which I assume is disease diagnosis, but it is not stated explicitly), describe how some medical imaging may be missing for certain patients (imputation), then explain that there may be noise across different medical imaging systems (UDA), then list the other applications where this arises with citations (e.g., These phenomena have also been documented in advertising applications [1], ...).
2. I was surprised by the difference between Adaptation-Partial and the other two train/test conditions in Figure 2 when p=30%. Out of curiosity, do the authors have an explanation for this discrepancy? I would have predicted that, if most of the information necessary for prediction was available in the remaining 70% of the image that the performance of these cases would be very similar.  I think it would be helpful to see the accuracy on the source domain and the labeled target domain to better understand that result.

**Experience Assessment:**

I have read many papers in this area.

**Review Assessment: Checking Correctness Of Derivations And Theory:**

I assessed the sensibility of the derivations and theory.

**Review Assessment: Checking Correctness Of Experiments:**

I assessed the sensibility of the experiments.

**Review Assessment: Thoroughness In Paper Reading:**

I read the paper at least twice and used my best judgement in assessing the paper.

---

> ### Author Response · Authors · 2019-11-13
> **Answer Reviewer 4 - Clarified joint domain shift and missing data for ads experiments (Part 2/2)**
>
> B) Other (*Additional notes.*)
>
> 1. We will update the paper in a revised version with the reviewer's recommendations for a more direct explanation of the motivations. We will as suggested rely on some examples from the literature such as the ones shortly described below, that address imputation. These methods do not perform adaptation.
>
> [Cai2018]: presents an application of missing data for multi-modality clinical applications, such as tumor detection and brain disease diagnosis. On these problems, different modalities can usually provide complementary information, which commonly leads to improved performance. However, some modalities are often missing for some subjects due to various technical and practical reasons and thus multi-modal data might be incomplete. This paper, which is the closest to ours in the literature, attempts at reconstructing a fixed missing modality on target subjects with direct supervision from fully observed training instances.
>
> [Tran2017]: addresses an object recognition problem with multi-modal sensors data. It is common on these applications that the sensing equipment experiences unforeseeable malfunction or configuration issues, leading to corrupted data with missing modalities. Most existing multi-modal learning algorithms could not handle missing modalities, and would discard either all modalities with missing values or all corrupted data. This paper relies on a stochastic distribution of the missing modalities while we deal with fixed missing modalities.
>
> [Wang2018]: presents a model for rating prediction tasks with multi-modal data and draws the link between cold-start and missing modality considering the same setting as [Tran2017] in the context of recommendation.
>
> 2. In the remaining non-missing 30% of the image there might still be important pixels (such as the top part of a "7" digit which could be mistaken for a "1" digit if not present) which can justify why the partial model is not performing better than the imputation model which could have imputed this missing information correctly. The difference with the missing model is probably due to suboptimal hyperparameters for the partial model as these were tuned for $p = 0.5$ and used as such for the other instances of $p$. Each of the three models should converge to the value of 77.6% (mentioned in Table 1 in the SVHN $\rightarrow$ MNIST Adaptation-Full line) when $p \rightarrow 0$.
>
> [Cai2018] Cai et al. Deep adversarial learning for multi-modality missing data completion. KDD 2018
> [Wang2018] Wang et al. LRMM: Learning to Recommend with Missing Modalities. EMNLP 2018
> [Tran2017] Tran et al. Missing Modalities Imputation via Cascaded Residual Autoencoder. CVPR 2017

---

> ### Author Response · Authors · 2019-11-13
> **Answer Reviewer 4 - Clarified joint domain shift and missing data for ads experiments (Part 1/2)**
>
> Thanks a lot for your feedback and your recommendations. We provide below a detailed response.
>
> A) Joint domain shift and missing data hypothesis for the ads experiments:
>
> We do agree that this is not obvious. As mentioned in the paper, the ads problem was our initial motivation for this work and our formulation of the problem comes from preliminary exploratory data analyses performed on ads datasets. We have added in Appendix E in the new paper version, distribution plots (Figure 6) and mean values (Table 6) for the different observed features used in the ads-kaggle dataset for the source and target domains. This shows that there is indeed a domain shift between the two domains. The same conclusion holds for the ads-real dataset. More details are provided below.
>
> Your description of the problem in the detailed comments (*Questions that impacted rating.*) is basically right. We figured out however that we might not have been precise enough in the text and we provide below more details clarifying the experimental setting.
>
> The source dataset is composed of all user-partner pairs for which the user visited the partner and the target dataset is composed of all the user-partner pairs for which the user never visited this partner. A key point here is that there are several partners (and of course users) per domain, and this was probably not clear enough from the text. Typically we could expect thousands of partners depending on the size of an ads company's partner portfolio. For the source domain, we have available complete data (mean statistics on all visited partners + traces on a specific ad partner for a user-partner pair) and for the target domain only partial data (mean statistics but no partner specific traces for a user-partner pair).
> Regarding domain shift, in Figure 6 Appendix E, we plot the normalized feature distributions for the source (blue plots) and target (red plots) domains. While some features have a similar distribution for the source and target domains, many have completely different distributions indicating a clear domain shift. This is synthesized in Table 6 Appendix E, giving the mean values for all the features for the two domains. We notice that feature 5 is missing on the target.
>
> This shift was initially a surprising finding for us too. Our hypothesis is that the source domain includes users with a higher overall activity both for visiting partner websites and for interacting with the websites. Target domain includes users that are probably less active. This is confirmed by the mean value of the features in the two domains in Table 6 Appendix E: feature distributions from users in the source domain tend to have higher mean values than features in the target domain. These features typically measure click, visit and sale activities which is consistent with the above hypothesis.

---

### Official Review · AnonReviewer2 · 2019-10-26
**Official Blind Review #2**

**Rating:** 3

**Review:**

This paper proposed to address a compound problem where missing data and distribution shift are both at play.  The paper goes on to describe some heuristic methods that resemble the gradient reversal methods due to Ganin et al for handling both problems.

The novel part of the paper over DANNs is the joint, end-to-end training of latent representations for missing data,  While it is sloppy with terminology, the paper is overall reasonably easy to follow although it might mislea a novice reader and sufficient details are provided to replicate their results.

The major problem here is the problem appears to be underspecified, and its not clear under what conditions if any the proposed methods are valid. Moreover it’s not clear to what extent the experimental results should ameliorate these concerns.

If the data is not missing at random then there is presumably confounding. The authors dance around this topic, just asserting that they are handling non-stochastic missing data but do not say precisely what is assumed about the relationship between the observed and missing data.

In short the paper addresses an under-specified problem with a heuristic technique based upon domain-adversarial nets which have recently been shown have a number of fundamental flaws. It's never made clear under what assumptions this proposed procedure is valid and the paper misrepresents the prior work on lable shift, including the theoretically sound work, e.g.:

"we assume covariate shift as in most UDA papers e.g. Ben-David et al. (2010); Ganin & Lempitsky (2015).”
>>>  Ben-David 2010 is not about covariate shift ….

Some minor thoughts:

“some components of the target data are systematically absent”
>>> 	Not clear what “component” means at this point

“We propose a way to impute non-stochastic missing data”
>>> 	What does this mean? Is non-stochastic, not missing at random? What is the pattern of missing-ness conditioned on? What assumption, if any, is made?

“This key property allows us to handle non-stochastic missing data,”
>>> 	again what precisely does this mean?

“Consider that x has two components (x_1, x_2)…”
>>>	sloppy  notation:
	“Source features” x_s = (x_S1, x_S2) are always available


I read the author's reply but do not believe that the responses are satisfactory. The authors do not address the primary concerns clearly and do not point to specific improvements in the draft that might cause me to change my mind.



**Experience Assessment:**

I have published in this field for several years.

**Review Assessment: Checking Correctness Of Derivations And Theory:**

I assessed the sensibility of the derivations and theory.

**Review Assessment: Checking Correctness Of Experiments:**

I assessed the sensibility of the experiments.

**Review Assessment: Thoroughness In Paper Reading:**

I read the paper at least twice and used my best judgement in assessing the paper.

---

> ### Author Response · Authors · 2019-11-13
> **Answer Reviewer 2 - Clarified problem specifications and missing data assumptions (Part 2/2)**
>
> B) Other points:
>
> * The original contribution is the joint adaptation-imputation model. In the paper we propose two instances of this model, one is based on adversarial training as you mention, the other one is based on distribution matching by optimal transport. Other mechanisms could have been used as well, but these two are representative of two state of the art families of adaptation methods. As indicated we found out that both methods lead to similar performance on our datasets, but each require a specific parameter tuning.
>
> * Concerning the validity of the method, we have no formal proof of consistency. However, the experiments show that under the hypotheses specified above, the method offers systematic improvement w.r.t. a selection of representative baselines and for  different instances of the adaptation-imputation problem. The results have been obtained on datasets with extremely different characteristics, including real world data with all their intrinsic complexity, and validated by intensive ablation studies. Of course, we do not claim to cover all situations, but we believe that our experiments already bring a strong evidence for the potential of the proposed method.
>
> * The reference to [Ben-David2010] as an example of domain shift was indeed an error and was removed. The work of [Ben-David2010] has however been fundamental for motivating and justifying several approaches to the adaptation problem. [Ganin2015] for example builds on the ideas introduced in [Ben-David2010] for justifying their approach and for approximating the H-divergence quantity introduced in [Ben-David2010]. Their conclusions are also valid for our ADV model.
>
> * We use the term component in its classical mathematical or physical acceptation. All the data we deal with are represented as vectors and a component of a vector is a feature or a subset of features of this vector.
>
> * Regarding the notation for referring to the input data $x$, we have consistently used a dash "-" to refer either to the Source or the Target domain and then used 1 or 2 in subscript to S or T as a way of differentiating components 1 (always observed) and 2 (observed only for source data) in a S or T instance.
>
>
> [Ganin2015] Ganin et al.  Unsupervised Domain Adaptation by Backpropagation. ICML 2015
> [Little2014] Little and Rubin. Statistical analysis with missing data, volume 333. John Wiley & Sons, 2014
> [Rubin1976] Rubin. Inference and Missing Data. Biometrika, 63, 581-592. 1976
> [Ben-David2010] Ben-David et al. A theory of learning from different domains. 2010

---

> ### Author Response · Authors · 2019-11-13
> **Answer Reviewer 2 - Clarified problem specifications and missing data assumptions (Part 1/2)**
>
> Thank you for your review. We have taken note of your comments and try to provide detailed answers to the points you raised:
>
> A) Problem specification:
>
> All the data considered in the paper are vectors. Let $x \in \mathbb{R}^n$ be a complete data vector and $m \in \{0,1\}^n$ a binary mask indicating which entries of $x$ are missing (1 for missing and 0 for observed). Given a dataset $X = (x_{(i)})_{i \in \{1, ..., N\}}$, we define the missingness indicator matrix as $M=(m_{(i)})_{i \in \{1, ..., N\}}$ where $N$ is the number of instances. In the following, we remove index $i$ for clarity. The hypotheses are the following:
>
> 1. Source domain data are fully observed.
>
> 2. Target domain data are partially observed and the missingness pattern is fixed: the indicator mask $m$ is the same for all the target domain data. In the paper when one refers to "non-stochastic missing data" we mean that the mask pattern is fixed for all target domain data. For simplicity, we have used the notations $x_S=(x_{S_1},x_{S_2})$ (source) and $x_T=(x_{T_1},x_{T_2})$ (target) with $x_{S_1}$ and $x_{T_1}$ corresponding to the mask positions with 0 value, i.e. features observed on both domains, and $x_{S_2}$ and $x_{T_2}$ corresponding to mask positions with value 1, i.e. missing features for the target domain.
> Please see the discussion below on the classical terminology used for missing data (item "Rubin's theory for missing data").
>
> 3. $x_{S_2}$ and $x_{T_2}$ contain some information not present in $x_{S_1}$ and $x_{T_1}$. This means that the feature values $x_{S_2}$ and $x_{T_2}$ cannot be predicted directly (e.g. through a regressor) from the features $x_{S_1}$ and $x_{T_1}$ respectively.
>
> 4. The distribution of features $x_{T_2}$ conditioned on the features $x_{T_1}$ could be inferred provided that we have some supervision for training the conditional distribution model; this is typically the case if for some $x_{T_1}$ observation, one also observes $x_{T_2}$, however such a supervision is not available for our problem. This is where adaptation comes into play. For inferring the conditional probability, the model  makes use of the source domain information for which this supervision is available while adapting to the target domain. Note that the model operates in a latent space and not in the original one. It does not attempt to reconstruct the true conditional distribution of $x_{T_2}$, but a conditional distribution in a latent space for a projection of $x_{T_2}$ denoted $\widetilde{x}_{T_2} $ in the paper.
>
> 5. Finally, we make the typical covariate shift assumption seen in several UDA papers, such as DANN [Ganin2015] to address an unsupervised classification adaptation setting.
>
> These hypotheses map to several real world problems as mentioned in this paper and the methods introduced show important improvement on our datasets based on these specifications.
>
> * Rubin's theory for missing data:
>
> The foundations of missing data theory were established by Rubin [Rubin1976] and his colleagues [Little2014]. We briefly introduce this formalism in order to put in evidence the specificity and originality of our problem. Rubin distinguishes between a missingness pattern, which describes which values are missing and observed in the data and the missingness mechanism, which represents the statistical relationship between the probability of missing data and the data variables. Let $m$ define as above a pattern of missing data; the missingness mechanism is characterized by the conditional distribution of $m$ given $x$, $p_{\phi}(m | x) ~\forall x \in X$ where $\phi$ denotes a vector of unknown parameters describing the relationship between the $m$ and $x$ variables. $\phi$ is known as the mechanism of missing data and provides the basis for distinguishing between three categories of missing data problems: Missing Completely at Random - MCAR ($\forall x \in X, p_{\phi}(m|x)=p_{\phi}(m)$), Missing At Random - MAR ($\forall x \in X, p_{\phi}(m|x)=p_{\phi}\left(m| x ^{\mathrm{obs}}\right)$ with $x^{obs}$ being the observed features), and Missing Not At Random - MNAR that covers all the other cases.
>
> Our missingness mechanism trivially corresponds to MCAR on the target domain. However, the key idea behind Rubin's theory is that missingness is a variable with a probability distribution. In our case, the missingness pattern is deterministic, not stochastic. The problem is then more difficult than classical MCAR problems and does not lead for example to classical maximum likelihood solutions exploited in the literature.

---

### Decision · Program_Chairs · 2019-12-19

**Decision:**

Reject

**Comment:**

This paper addresses the problem of performing unsupervised domain adaptation when some target domain data is missing is a potentially non-stochastic way. The proposed solution consists of applying a version of domain adversarial learning for adaptation together with an MSE based imputation loss learned using complete source data. The method is evaluated on both the standard digit recognition datasets and a real-world advertising dataset.

The reviewers had mixed recommendations for this work, with two recommending weak reject and one recommending acceptance. The key positive point from R3 who recommended acceptance was that this work addresses a new problem statement which may be of practical importance. The other two reviewers expressed concerns over the contribution of the work and the validity of the problem setting. Namely, both R2 and R4 had significant confusion over the problem specification and/or under what conditions the proposed setting is valid.

It is a difficult decision for this paper as there is a core disagreement between the reviewers. All reviewers seem to agree that the proposed solution is a combination of prior methods in a new way to address the specific problem setting of this work.  However, the reviewers differ in precisely whether they determine the proposed problem setting to be valid and justified. Due to this discrepancy, the AC does not recommend acceptance at this time. If the core contribution is to be an application of existing techniques to a new problem statement than that should be clarified and motivated further.